# Communication and quorum sensing in non-living mimics of eukaryotic cells

Henrike Niederholtmeyer [1], Cynthia Chaggan[1] & Neal K. Devaraj[1]

Cells in tissues or biofilms communicate with one another through chemical and mechanical signals to coordinate collective behaviors. Non-living cell mimics provide simplified models of natural systems; however, it has remained challenging to implement communication capabilities comparable to living cells. Here we present a porous artificial cell-mimic containing a nucleus-like DNA-hydrogel compartment that is able to express and display proteins, and communicate with neighboring cell-mimics through diffusive protein signals. We show that communication between cell-mimics allows distribution of tasks, quorum sensing, and cellular differentiation according to local environment. Cell-mimics can be manufactured in large quantities, easily stored, chemically modified, and spatially organized into diffusively connected tissue-like arrangements, offering a means for studying communication in large ensembles of artificial cells.

---

[1] Department of Chemistry and Biochemistry, University of California, San Diego, La Jolla CA 92093, USA. Correspondence and requests for materials should be addressed to N.K.D. (email: ndevaraj@ucsd.edu)

In communities of single-celled and multicellular organisms, cell–cell communication enables cells to organize in space, distribute tasks, and to coordinate collective responses. Synthetic biologists have engineered living, communicating cells to form cellular patterns[1,2] and synchronize gene expression[3] but living systems are inherently challenging to study and engineer. Chemically constructed cell-mimics, as non-living, biochemically simplified and engineerable systems, could serve as models to study mechanisms of pattern formation and collective responses, and lead to the development of novel sensors and self-organizing materials. Important biochemical processes like protein synthesis[4,5], DNA replication[6], metabolism[7], and cytoskeletal functions[8] have been reconstituted and studied in single synthetic cell-mimics. While biochemical reactions in microfluidic chambers[9–11], in droplets[12,13] and on beads[14] can emulate aspects of intercellular communication, studies on systems that structurally resemble natural cells with their semi-permeable membranes have been limited in scope by the availability of communication channels and assembly methods. Addressing the scalable assembly of artificial cells, microfluidic methods have been developed to mass-produce highly homogeneous populations of phospholipid vesicles encapsulating active biomolecules[15–18]. Recent studies have demonstrated communication between synthetic microcompartments to induce gene expression[5,13,19,20] or chemical reactions[21–23] using small molecule signals. To implement communication, signaling molecules must travel between compartments. Some small molecules diffuse freely between compartments[5,13,19–22], phospholipid vesicles can be permeabilized by inserting alpha-hemolysin pores[5,23], and other synthetic microcompartments such as gel-shell beads[24], polymersomes[21], proteinosomes[23], and colloidosomes[22] can be assembled with permeable membranes. Signaling molecules for communication between artificial cell-mimics have so far been limited to small molecules. In contrast, signaling in multicellular organisms often involves secretion of proteins serving as growth factors or morphogens that provide cells with the information they need to develop into functional tissues[25].

Here, we aim to expand the communication capabilities of artificial cells by developing a cellular mimic that produces and releases diffusive protein signals that travel in and get interpreted by large populations of cell-mimics. We describe the microfluidic production of cell-mimics with a porous polymer membrane containing an artificial hydrogel compartment, which resembles a eukaryotic cell's nucleus in that it contains the cell-mimics' genetic material for protein synthesis and can sequester transcription factors. Cell-mimics are able to communicate through diffusive protein signals, activate gene expression in neighboring cell-mimics, and display collective responses to cell-mimic density similar to bacterial quorum sensing.

## Results

**Porous cell-mimics containing artificial nuclei.** We prepared porous cell-mimics capable of gene expression and communication via diffusive protein signals using a microfluidic method (Fig. 1a, b). First, water-in-oil-in-water double emulsion droplets were formed in a polydimethylsiloxane (PDMS) device (Supplementary Figure 1, Supplementary Movie 1). The droplets had a middle organic phase consisting of a 1-decanol and acrylate monomer solution and encapsulated DNA and clay minerals. Second, double emulsion droplets were collected and polymerized using UV light, inducing a phase separation of the inert 1-decanol to form porous microcapsules[26]. Third, following polymerization, we simultaneously permeabilized the polymer membrane and induced formation of a clay-hydrogel in their interior by adding a solution of ethanol and HEPES buffer. Membrane pores had diameters of 200–300 nm (Fig. 1a, Supplementary Figure 2). Polymer membranes were permeable to macromolecules up to 2 MDa but excluded 220 nm nanoparticles from about 90% of the microcapsules (Supplementary Figure 3). Like in similarly prepared porous microcapsules[26,27], polymer membranes were mechanically stable and rigid. Microcapsules could be centrifuged at high speeds, and only broke under high stress from a razor blade (Supplementary Figure 2). The encapsulated clay minerals

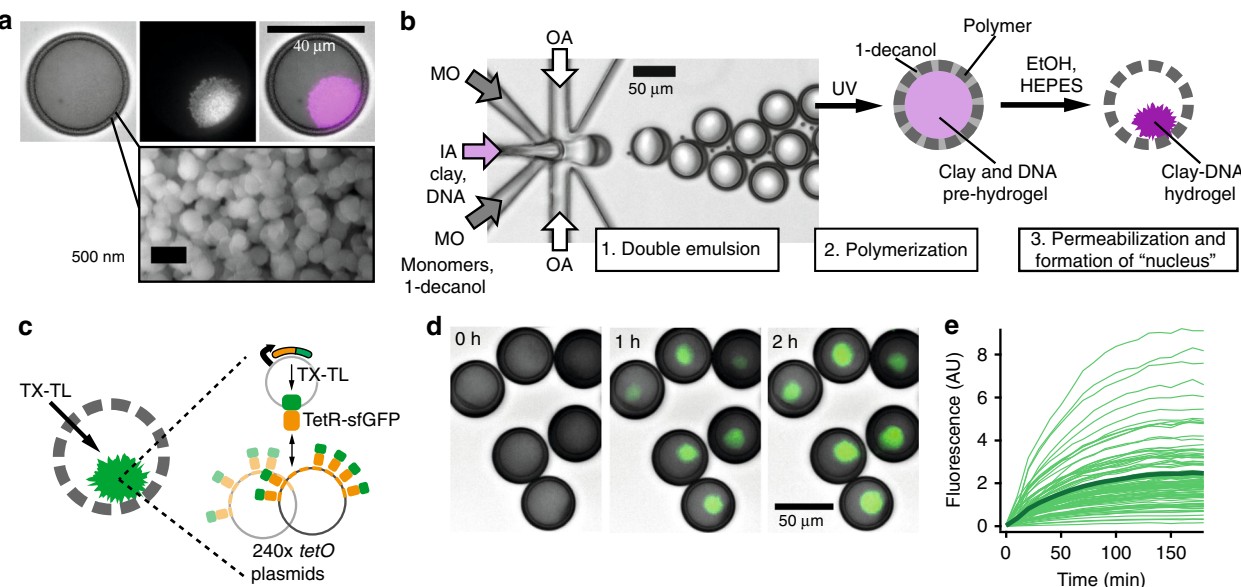

**Fig. 1** Formation of cell-mimics with artificial nuclei capable of gene expression. **a** Optical micrographs (top) of a cell-mimic with GelRed stained hydrogel nucleus (brightfield, red fluorescence, merge) and scanning electron microscopy of porous cell-mimic membrane (bottom). **b** Microfluidic production of double emulsion droplets encapsulating a pre-hydrogel in a photocurable middle layer, and schematic of subsequent processing steps. IA: Inner aqueous, MO: middle organic, OA: Outer aqueous phase. **c** Schematic and timelapse images **d** of expression and capture of TetR-sfGFP in hydrogel nuclei (green, merged with brightfield images). TX-TL reagents were added at 0 h. **e** Dynamics of fluorescence signal increase in the hydrogel nuclei of 100 cell-mimics with average shown in bold

had a large capacity for binding and capturing DNA from solution, and retained DNA in the clay-DNA hydrogel that formed after electrolyte addition (Supplementary Figure 4)[28]. During permeabilization of the polymer membrane with a solution of HEPES buffer and ethanol, the encapsulated clay minerals and DNA condensed into a round hydrogel structure in the microcapsules' interior that is analogous to the cell nucleus (Fig. 1c). In at least 95% of microcapsules, only one clay-DNA hydrogel nucleus formed that measured approximately half of the microcapsules' diameter in size. In solution, not templated by a surrounding microcapsule, clay minerals form irregularly shaped and sized hydrogel particles (Supplementary Figure 5). DNA adsorbs to the surfaces of clay minerals through electrostatic interactions, cation bridges, hydrogen bonding and ligand exchange[28–30]. Clay-DNA hydrogel nuclei had a porous structure, allowing macromolecules of up to 500 kDa to diffuse into the hydrogel while 2000 kDa dextran was partially excluded from hydrogel nuclei (Supplementary Figure 3). The permeability of the hydrogel nuclei to large macromolecules suggested that the DNA would be accessible to the transcription machinery. As a final step in the preparation of cell-mimics, we passivated their polymer membrane with polyethylene glycol (PEG) to prevent non-specific binding of proteins (Supplementary Figure 6).

**Gene expression in cell-mimics**. The porous structure of the polymer membrane allowed supply of cell-free transcription and translation (TX-TL) reagents from the outside to induce synthesis of proteins encoded by the DNA in the cell-mimics' hydrogel nuclei. Even ribosomes, the largest components of TX-TL reagents, were able to diffuse into cell-mimics through their porous membranes (Supplementary Figure 7). To capture protein products within cell-mimics, we expressed a fusion protein of the tetracycline repressor TetR and sfGFP (TetR-sfGFP) as a fluorescent reporter. TetR binds the *tet* operator sequence (*tetO*). A co-encapsulated 240 × *tetO* array plasmid localized the reporter protein to the hydrogel nucleus (Fig. 1c), which increased in fluorescence after TX-TL addition (Fig. 1d, Supplementary Movie 2). Localization of TetR-sfGFP to the hydrogel nucleus was reversible and due to the specific interaction of TetR with *tetO* sites. Addition of anhydrotetracyline, which prevents TetR from binding DNA, caused a substantial unbinding of TetR-sfGFP. Without the *tetO* plasmid, fluorescence increased in solution but not in hydrogel nuclei (Supplementary Figure 8). In *tetR*-sfGFP/ *tetO* cell-mimics, fluorescence increased substantially in almost all cell-mimics (Fig. 1e). Variations in intensity were likely due to differences in DNA capture during formation of hydrogel nuclei (Supplementary Fig. 5). Cell-mimics retained full expression capabilities after 2 years of storage, and separate batches showed comparable expression levels and dynamics (Supplementary Figure 9).

**Protein exchange between cell-mimics**. Due to their porosity, cell-mimics likely released mRNA and protein products that diffused into neighboring cell-mimics. To demonstrate that neighboring cell-mimics exchanged protein products with each other, we prepared sender cell-mimics, fluorescently labeled in their polymeric membranes and containing the *tetR*-sfGFP expression plasmid, and receiver cell-mimics containing the *tetO* array plasmid to capture the reporter protein. When both cell-mimic types were mixed at approximately a one to one ratio, only the nuclei of the receiver cell-mimics increased in fluorescence (Fig. 2a). To explore how far TetR-sfGFP protein originating from a given sender cell-mimic travelled, we used a large excess of receivers and spread them densely into a circular 3.5 mm wide colony. Under these conditions, TetR-sfGFP spread from sender

to surrounding receiver cell-mimics but stayed localized in patches around individual sender cell-mimics (Fig. 2b, Supplementary Figure 10a). This pattern of captured protein around source cell-mimics persisted for 24 h after expression ended, demonstrating that TetR-sfGFP was essentially trapped in the hydrogel nuclei once it was bound in the high local density of *tetO* sites. Assuming free diffusion, we would expect protein gradients to have disappeared within 5 h in similar geometries (Supplementary Figure 10b–d).

To test the preference of a given cell-mimic to bind protein originating from its own DNA, we prepared *tetR*-mCherry / *tetO* cell-mimics that accumulated red fluorescence in their hydrogel nuclei (Supplementary Fig. 10). When mixed with *tetR*-sfGFP / *tetO* cell-mimics (Fig. 1c), there was essentially no difference in relative fluorescence in either channel between the cell-mimic types, indicating that in close proximity, neighboring cell-mimics completely exchanged protein products (Supplementary Figure 11). While transcription occurred in the hydrogel nuclei where DNA was localized, these results indicate that translation was likely not localized to the cell-mimic a given mRNA originated from. However, because mRNA lifetime in TX-TL reagents is short, and mRNA thus has a limited diffusion range, we expected the localization of TetR-sfGFP and TetR-mCherry to depend strongly on distance between cell-mimics. We distributed the two cell-mimic types in a reaction chamber so that they mixed in the center but remained separate on either side. Cell-mimics in the center showed mixed fluorescence while cell-mimics on the sides fluoresced primarily in one channel (Fig. 2c), demonstrating that locally, on the order of few cell-mimic lengths, proteins exchanged with little hindrance by the polymer membranes, whereas exchange of protein with distant cell-mimics was limited by diffusion.

**Communication through a diffusive transcriptional activator**. Communication in vesicle-based cell-mimics has so far been limited to small molecule signals such as quorum sensing molecules[19,20] or IPTG and glucose, combined with membrane pores, like alpha-hemolysin[5,23]. Our porous cell-mimics exchanged proteins with their neighbors, suggesting they are able to communicate with each other directly through genetic regulators. To demonstrate this we constructed a two-stage activation cascade and distributed the network into two separate cell-mimic types. T3 RNA polymerase (T3 RNAP) served as a diffusive signaling molecule transmitting the instruction to express a reporter gene from activator to reporter cell-mimics. Activator cell-mimics contained the template for the expression of T3 RNAP. Reporter cell-mimics contained the template for the T3 RNAP-driven synthesis of the TetR-sfGFP reporter as well as *tetO* array plasmids to capture the reporter protein. When both cell-mimic types were mixed, reporter cell-mimics expressed and bound the fluorescent reporter (Fig. 3, Supplementary Movie 3), while activator cell-mimics alone did not increase in fluorescence (Supplementary Figure 12).

**Artificial quorum sensing**. We hypothesized that T3 RNAP could serve as a soluble signaling molecule providing cell-mimics with information about population density. Indeed, cell-mimics containing both the activation circuit and reporter constructs (Fig. 4a) underwent a collective response where fluorescence accumulated in cell-mimics only at high densities. At low cell-mimic densities, signals from the hydrogel nuclei were not detectably different from background fluorescence (Fig. 4b). We titrated the density of cell-mimics in a fixed volume and found a sharp transition from off to on, which resembled bacterial quorum sensing responses to cell density[31] even though, unlike in

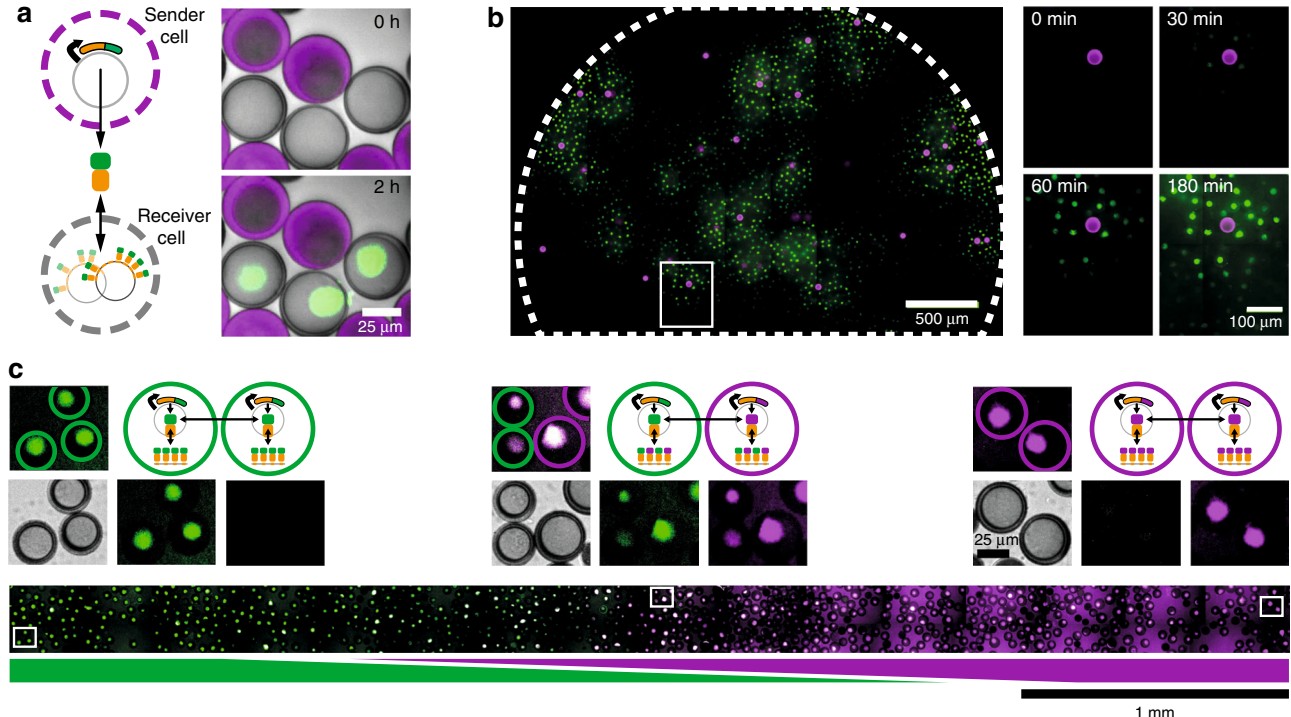

**Fig. 2** Protein exchange between neighboring and distant cell-mimics. **a** Schematic of diffusive TetR-sfGFP exchange between neighboring sender and receiver cell-mimics and timelapse images of neighboring senders (rhodamine B stained membranes) and receivers (unstained membranes). Merge of brightfield and fluorescence channels (sender membranes, magenta; TetR-sfGFP, green). **b** Distribution of TetR-sfGFP (green) in a dense droplet of receivers and sparse senders (magenta) after 3 h of expression. A small region around a sender (white box) is magnified and spreading of fluorescence is shown at different time points. **c** Inhomogeneous mix of two types of cell-mimics producing and binding different color reporter proteins. tetR-sfGFP / tetO (green) and tetR-mCherry / tetO cell-mimics (magenta) were distributed in a channel to stay separate at the sides and mix in the center. Bottom image shows the distribution of sfGFP and mCherry fluorescence after 5 h. Merge of the two channels results in a white signal (middle). Magnified images from indicated positions along the channel are shown above. Merged image with cell-mimic types indicated by colored, dashed circles (top), and brightfield, sfGFP and mCherry signals shown separately (below)

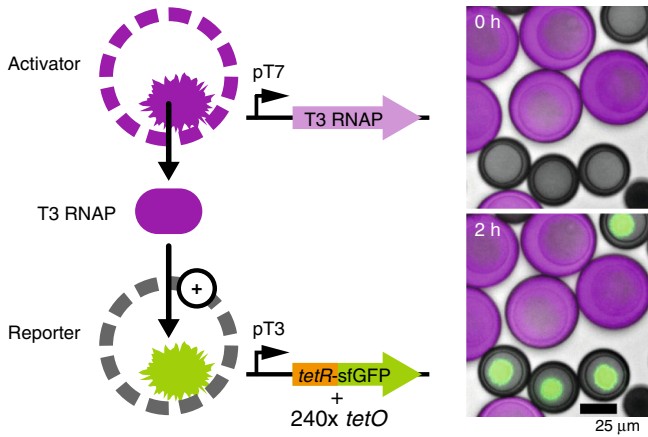

**Fig. 3** Communication between cell-mimics via a diffusive genetic activator. Schematic of the two types of cell-mimics communicating through a distributed genetic activation cascade. Micrographs show a merge of brightfield images with rhodamine B fluorescence in the membranes of activators (magenta) and fluorescence of TetR-sfGFP (green) in the hydrogel nuclei of reporters directly after addition of TX-TL and after 2 h of expression

bacterial quorum sensing mechanisms, the T3 activation circuit did not contain positive feedback. The threshold cell-mimic density at which expression of the reporter turned on was 400 cell-mimics in 4.5 µl TX-TL (Fig. 4c). Cell-mimics that

constitutively expressed the reporter (Fig. 1c) accumulated fluorescence in their hydrogel nuclei regardless of their density (Supplementary Figure 13). The collective response to density can be explained by T3 RNAP release from cell-mimics. At low densities, T3 RNAP is diluted in the comparably large volume of the sample, while at high density a sufficient concentration of transcriptional activator accumulates to turn on expression of the reporter. Titrating the T3 RNAP template DNA in TX-TL reactions, we found a steep transition from low to high expression with a half-maximal activation at 10 pM (Supplementary Figure 14). The calculated bulk concentration of T3 RNAP template in an artificial quorum sensing experiment at the threshold density of 400 cell-mimics per droplet is 12.5 pM, similar to the activation threshold in bulk solution.

During development, cells interpret signals secreted by their neighbors to differentiate into specialized cell-types that express different sets of genes[25]. We aimed to emulate cellular differentiation according to local environment by combining the artificial quorum sensing network with a constitutively expressed tetR-mCherry reporter that turns on irrespective of cell-mimic density (Fig. 4d). We distributed cell-mimics unevenly in a long narrow reaction chamber (Fig. 4e), and analyzed the fluorescence of individual hydrogel nuclei according to their location in the density gradient. While absolute fluorescence intensities and background fluorescence increased with cell-mimic density, hydrogel nuclei from the high density area displayed visibly higher sfGFP:mCherry ratios than hydrogel nuclei in the dilute region that primarily displayed mCherry fluorescence (Fig. 4f). In the continuous density gradient, we

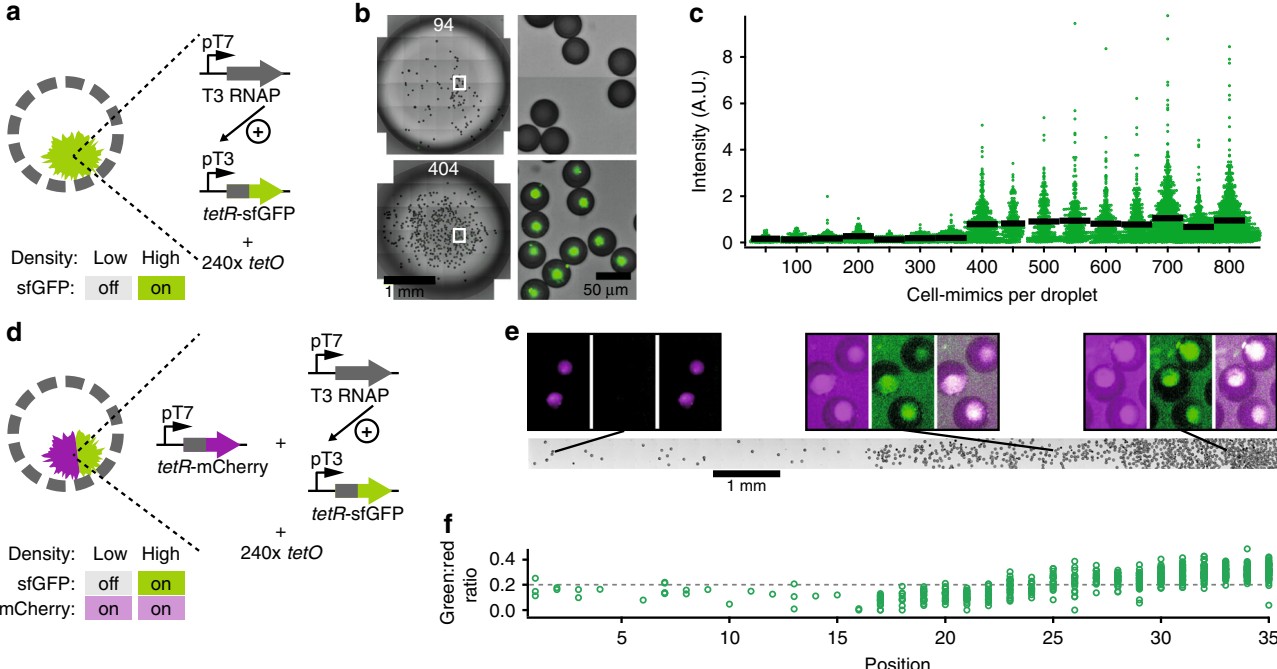

**Fig. 4** Density sensing in populations of cell-mimics. **a** Artificial quorum sensing cell-mimics contain T3 activation cascade DNA templates and 240 × *tetO* plasmids. **b** Micrographs of cell-mimics in 4.5 µl droplets of TX-TL (left). The number of cell-mimics is indicated. Enlarged regions (indicated by white boxes) show presence and absence of fluorescence (green) in hydrogel nuclei after 3 h of expression. **c** Scatter dot plot of fluorescence intensities in individual cell-mimics at different densities. Each density category combines data in increments of 50 cell-mimics per droplet and contains data from at least 156 cell-mimics (Methods). Black bars show average fluorescence. **d** A 2-color response to density is achieved by adding a constitutively expressed reporter (pT7-*tetR*-mCherry), which is on independent of density. **e** 2-color density sensors were spread at increasing density in an elongated chamber (brightfield image, bottom). Panels above show magnified fluorescence images of indicated regions (mCherry fluorescence: magenta, left; sfGFP fluorescence: green, middle; merge of fluorescence channels: right). Images in each channel are window leveled to the same values and have a width of 70 µm. **f** Ratio between sfGFP to mCherry fluorescence in individual hydrogel nuclei along the chamber. Positions correspond to tile regions of the image above

observed some graded responses in the center of the chamber at medium density. However, plotting sfGFP against mCherry fluorescence for individual hydrogel nuclei revealed two distinct populations of fluorescence signals according to position in the chamber (Supplementary Figure 15).

## Discussion

We developed porous cell-mimics capable of gene expression and communication via diffusive protein signals. The clay-DNA hydrogel in the cell-mimic's interior resembles a eukaryotic cell's nucleus, and represents a different strategy to compartmentalize artificial cells[17,32,33]. Our study demonstrates that clay minerals and clay-DNA hydrogels are useful hybrid materials for synthetic biology and the assembly of artificial cell-mimics. In fact, clay minerals have been proposed as favorable environments for prebiotic evolution because a wide variety of organic molecules adsorbs to their surface. Clay minerals have been shown to possess catalytic properties[34] and to enhance protein synthesis in cell-free expression systems[28]. Unlike lipid vesicles that require careful matching of osmolarities and gentle separation techniques, the cell-mimics reported here with their porous polymer membranes were physically highly stable, easily transferred into new media by centrifugation, and retained full expression capabilities after 2 years of storage. Microfluidic production of highly homogeneous cell-mimics will facilitate studies requiring large quantities of cell-mimics as our results on spatially arrayed, communicating cell-mimics demonstrate. Passivating cell-mimics' polymer membranes with PEG, we showed that their membranes could be chemically modified, which will allow

further functionalization, for example, for immobilization on substrates, to target specific proteins to the membrane or to tune membrane permeability.

So far, communication in synthetic, non-living cell-mimics has been limited to small molecule signals[5,19,20,22,23]. The porous cell-mimics developed here expand the communication capabilities of artificial cells to large macromolecules like RNAs and proteins. We showed that genetic circuits could be distributed into separate cell-mimics, which allowed them to share tasks. Such modularity might facilitate the assembly of functional, distributed gene circuits by titrating cell-mimics containing different parts of a network. Furthermore, individual components of circuits located in different cell-mimics can be spatially organized to generate spatiotemporal expression patterns. Protein signals play an important role for cell–cell communication in multicellular organisms, where cells release and receive protein signals in the form of hormones, growth factors, and morphogens[25]. We believe that reaction–diffusion models explaining developmental processes in multicellular organisms may be tested and emulated in artificial, tissue-like arrangements of cell-mimics even when they are composed of unnatural materials, operate on different scales and with different molecular mechanisms. Clearly, a major difference between an artificial tissue assembled from porous cell-mimics and natural tissues is that translation of proteins takes place both inside and outside of the cell-mimic a given mRNA originates from, and that mRNAs and proteins can freely diffuse. In this regard, our system emulates features of the syncytium stage during *Drosophila* embryogenesis, when thousands of nuclei accumulate in the unseparated cytoplasm of the oocyte[35]. In another analogy to this developmental stage, we showed trapping

of the TetR transcription factor in nuclei containing binding sites. Nuclear trapping is a mechanism that is responsible for establishing sharp gradients of phosphorylated ERK/MAPK (dpERK) across the syncytical *Drosophila* embryo by limiting diffusion of dpERK[36]. In future work, diffusion rates of proteins through tissue-like arrangements of cell-mimics may be tuned by using nuclear binding sites that restrict diffusion of DNA binding proteins or by modifying polymer membrane permeability.

A genetic activation circuit led to a remarkable collective response to cell-mimic density, which resembled bacterial quorum sensing. In contrast to bacterial quorum sensing[31], the mechanism of our artificial quorum sensing involved no positive feedback loop and employed a protein instead of a small molecule, showing that artificial cells can emulate biological phenomena using unnatural parts and mechanisms. Several molecular mechanisms can produce cooperative or ultrasensitive behaviors[37]. Polyvalent interactions enhance functional affinities between binding partners by favoring ligand rebinding after an initial binding event occurred[37–39]. We believe the switch-like behavior of the artificial quorum sensing response is due to cooperative binding of the T3 RNAP to highly polyvalent hydrogel nuclei containing high local concentrations of T3 RNAP promoters, combined with large distances between cell-mimics at low densities (Supplementary Figure 16), and an already sigmoidal response curve to T3 RNAP template concentration (Supplementary Figure 14). Collective responses can lead to greater accuracy and reduce noise[3], which will be particularly helpful for the assembly of reliably functioning cell-mimics, which often suffer from variability in gene expression[5,40].

In conclusion, our system has a number of potential uses, including programming cell-mimics to collectively sense and respond to their environment. Indeed, artificial cell-mimics could be used to develop sensors and self-organizing materials, as well as being arrayed into synthetic tissues of artificial cells, which could serve as simplified models for reaction–diffusion processes.

## Methods

**Fabrication of microfluidic chips**. PDMS devices were prepared by standard soft lithography methods to produce devices with a design as shown in Supplementary Figure 1. A silicon wafer patterned with SU-8 photoresist served as a mold for PDMS devices. The mold was prepared following standard photolithography procedures to produce a feature height of 43 µm. Design of the flow-focusing junction was adapted from Desphande et al.[16] (Supplementary Figure 1). PDMS (Sylgard, Dow Corning) was prepared at a 1:10 ratio and cured for 1 h at 80 °C. PDMS devices were bonded to PDMS spincoated cover glass using oxygen plasma (50 watt for 30 s at 0.45 torr). After bonding, devices were baked overnight at 120 °C to recover hydrophobicity. Channels downstream of the flow-focusing junction were rendered hydrophilic by a treatment with a 5% (wt/vol) polyvinyl alcohol (PVA) solution[16], which was flowed in the inlet for the outer aqueous solution for 5 min while blowing air into the other inlets to avoid contact of the PVA-solution with device regions upstream of the flow-focusing junction. The remaining PVA-solution was removed by applying vacuum to the outlet. Following PVA-treatment devices were baked again for 2 h at 120 °C and used immediately or stored for up to 2 months.

**Production of porous cell-mimics with DNA-hydrogel nuclei**. 2% (wt/vol) Laponite XLG (BYK Additives) clay stock was prepared by mixing 10 ml of ultrapure $H_2O$ on a magnetic stir plate to create a vortex. 200 mg of Laponite XLG were slowly added into the vortex and left to stir for 2 h until clear. The dispersion was then stored at 4 °C and used for up to a week. Photoinitiator 2,2-dimethoxy-2-phenylacetophenone was dissolved at 5% (wt/vol) in 1-decanol and in Trimethylolpropane ethoxylate triacrylate (ETPTA, Sigma-Aldrich, Mn 428). ETPTA with photoinitiator was stored at 4 °C and used for up to a week. Double emulsion droplets were prepared with an inner aqueous solution (IA) containing 0.4% (wt/vol) laponite XLG, 15% (vol/vol) glycerol, 50 mg/ml poloxamer 188, 20 µM sulfo-Cy5 and up to 300 ng/µl plasmid or linear DNA. The middle organic phase (MO) was composed of glycidyl methacrylate (GMA, Sigma-Aldrich), ETPTA, and 1-decanol at a 48:32:20 ratio and contained 2.6% (wt/vol) photoinitiator and 0.25% (vol/vol) Span-80 to produce porous microcapsules[26]. For fluorescently labeled microcapsule membranes, the MO phase contained 0.1 mg/ml methacryloxyethyl

thiocarbamoyl rhodamine B. The outer aqueous phase (OA) was 15% (vol/vol) glycerol with 50 mg/ml poloxamer 188.

Using syringe pumps, the three phases were flowed through the microfluidic device at speeds of 3 to 12 µl/h for the IA, 30 to 70 µl/h for the MO and 250 to 500 µl/h for the OA phase. Flow rates were adjusted to produce a stable formation of double emulsion droplets and then left unchanged for collection of droplets. Typically, about 200 µl of double emulsion were collected from the chip. The emulsion was then placed in a 2 mm thick chamber built from cover glass and exposed to 350 nm UV light for 30 s using a UV reactor (Rayonet). The dispersion of polymerized microcapsules was then added to 2 ml solution of 70% Ethanol containing 200 mM HEPES pH 8 to permeabilize the shell and to form the DNA-clay-hydrogel nucleus. This stock was stored at −20 °C until use.

To prevent non-specific binding of proteins to porous polymer membranes, microcapsules were treated with polyethylene glycol (PEG). We coupled amino-PEG12-alcohol to the epoxide functionalities on the polymer shells. First, microcapsules were washed with 200 mM sodium carbonate buffer pH 10 by centrifugation. All supernatant was removed from the capsule pellet and a solution of 250 mM amino-PEG12-alcohol in 50% ethanol pH 10 was added to the pellet. Microcapsules were incubated at 37 °C for reaction overnight and then washed with 100 mM HEPES pH 8. These PEGylated cell-mimics were either used directly or stored in 70% ethanol 200 mM HEPES pH 8 at −20 °C.

**DNA templates**. Plasmids used in this study are listed in Supplementary Table 1. Plasmid DNA was purified using the NucleoBond Xtra Midi kit (Macherey-Nagel), followed by an isopropanol precipitation and resuspension of the DNA pellet in ultrapure $H_2O$ to prepare highly concentrated plasmid stocks and maximize expression in TX-TL reactions. The template for T3 RNA polymerase was on linear DNA prepared by PCR from a plasmid template. DNA template concentrations encapsulated in cell-mimics are listed in Supplementary Table 2.

**Cell-free transcription and translation reactions**. *E. coli* lysate for TX-TL reactions was prepared by freeze-thawing from *E. coli* BL21-Gold (DE3)/pAD-LyseR[41]. To induce T7 RNA polymerase activity besides *E. coli* RNA polymerase activity the main culture was induced with 0.5 mM IPTG an optical density measured at 600 nm ($OD_{600}$) of 0.5–0.6. Cells were harvested by centrifugation at an $OD_{600}$ of 1.4 and resuspended in ice cold S30A buffer (14 mM magnesium glutamate, 60 mM potassium glutamate, 50 mM Tris, pH 7.7). Cells were again pelleted by centrifugation, and after determining pellet mass, cells were resuspended in two volumes (relative to cell mass) of S30A buffer containing 2 mM DTT and frozen at −80 °C. For lysate preparation, cells were thawed, vigorously vortex mixed for 3 min and incubated on an orbital shaker at 300 rpm at 37 °C for 45 min. Cell suspensions were again vortex mixed and incubated for another 45 min under shaking at 37 °C. Following incubation, cells were again vigorously mixed and the lysate was cleared by centrifugation at 50,000×*g* for 45 min. Cell debris-free, clear supernatant was collected and frozen in aliquots at −80 °C until use. For the final composition of TX-TL reactions, cell lysate was diluted 2.5-fold with reaction buffer, microcapsules or DNA, and other additions as needed. This resulted in the following concentrations in the TX-TL reaction: 4.6 mg/ml protein, 7 mM Mg-glutamate, 60 mM K-glutamate, 3.5 mM DTT, 0.75 mM each amino acid except leucine, 0.63 mM leucine, 50 mM HEPES, 1.5 mM ATP and GTP, 0.9 mM CTP and UTP, 0.2 mg/ml tRNA, 0.26 mM CoA, 0.33 mM NAD, 0.75 mM cAMP, 0.068 mM folinic acid, 1 mM spermidine, 30 mM 3-PGA, 3.5% PEG-8000. When linear DNA templates were used, they were stabilized by adding 4 µM of chi6 duplex DNA to the TX-TL reaction[42]. TX-TL reactions were incubated at 29 °C for expression.

**Gene expression in cell-mimics**. For cell-mimics, expression reactions typically consisted of 1 µl concentrated cell-mimics in 100 mM HEPES pH 8 and TX-TL reagents for a final volume of 5 µl. Droplets of 4.5 µl of this mixture were pipetted onto a 35 mm Lumox dish (Sarstedt). The gas permeable substrate ensured homogeneous sfGFP expression in the sample. The cell-mimic droplet was covered with cover glass and sealed with a ring of vacuum grease to prevent evaporation and provide a spacer. The reaction volume was scaled up for experiments in larger samples, and was 20 µl in Fig. 2b and 35 µl for long, narrow reaction chambers in Figs. 2c and 4e. Long, narrow reaction chambers were made from two parallel 20 mm lines of vacuum grease with a gap of 2 mm, which was filled with TX-TL and cell-mimics and then sealed with cover glass.

**Preparation of labeled ribosomes**. *E. coli* ribosomes (New England Biolabs) were incubated in labeling buffer (50 mM HEPES pH 8.2, 100 mM KCl, 10 mM magnesium acetate) with a molar excess of Alexa Fluor 488 NHS Ester (ThermoFisher Scientific) for 90 min at room temperature. Free dye was removed by washing with labeling buffer in centrifugal filter devices with a molecular weight cut-off of 100 kDa. Each ribosome contained approximately eleven Alexa Fluor 488 labels.

**Plate reader reactions**. Plate reader reactions were performed in 384-well plates using a 10 µl reaction volume covered with 10 µl light mineral oil in a Tecan infinite F200 plate reader. GFP fluorescence was read every 5 min using a 485 nm ± 20 nm excitation filter and a 550 nm ± 35 nm emission filter, followed by 1 min of shaking.

Fluorescence intensity measurements were calibrated using purified sfGFP-His$_6$ to determine absolute concentrations.

**Imaging and image analysis**. Images were acquired using a spinning disk confocal microscope consisting of a Yokagawa spinning disk system (Yokagawa, Japan) built around an Axio Observer Z1 motorized inverted microscope (Carl Zeiss Microscopy GmbH, Germany) with a 20 × 1.42 NA objective. Large regions were imaged as tiles and stitched using ZEN Blue software. Further image processing and analyses were done in Fiji/ImageJ[43]. Fluorescence traces or endpoint intensities of individual hydrogel nuclei were extracted from timelapse movies by measuring fluorescence in manually selected oval regions in nuclei, using a non-fluorescent region in each cell-mimic for background subtraction. Artificial quorum sensing data was analyzed using the colony counter plugin in Fiji/ImageJ to segment and count cell-mimics and polymer beads in the stitched brightfield image of a droplet. When necessary, regions of interest were manually added or deleted. Fluorescence values of individual cell-mimics were mean fluorescence values of the individual segmented cell-mimics. For background correction, fluorescence values of segmented regions from each droplet were sorted. The lowest fluorescence intensities were from solid polymer beads, and we used this property for background correction. We removed the lowest 34% of values, which was the percentage of polymer beads in the sample, and used the highest of the removed values for background subtraction of the reduced list. Experiments (droplets with different amounts of cell-mimics) were performed for densities between 25 and 800 cell-mimics per droplet, which were binned every 50 cell-mimics. Each bin contained data from at least 156 analyzed cell-mimics.

Scanning electron microscopy was performed with a Zeiss SIGMA VP field emission scanning electron microscope using air-dried cell-mimics. To image cross sections of microcapsule polymer membranes, cell-mimics were cut using a razor blade.

## Data availability

The authors declare that all relevant data supporting the findings of this study are available within the paper and its Supplementary information files. Additional data are available from the corresponding author upon request.

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

## Acknowledgements

This work was supported by the Department of Defense (Army Research Office) through the Multidisciplinary University Research Initiative (MURI), award W911NF-13-1-0383. H. N. was supported by Early and Advanced Postdoc.Mobility fellowships from the Swiss National Science Foundation (P2ELP3_161844, P300PA_174346). We thank Lev Tsimring, Partho Ghosh and Andrew Rudd for helpful comments on the manuscript, Ahanjit Bhattacharya for labeled ribosomes, Andriy Didovyk for *E. coli* strain BL21-Gold (DE3)/pAD-LyseR, and Prof. Jeff Hasty and Ryan Johnson for their collaboration in

microfluidic chip fabrication. We thank the UCSD School of Medicine Microscopy Core with grant NS047101 and Jennifer Santini for assistance with scanning electron microscopy, and the Waitt Advanced Biophotonics Core Facility of the Salk Institute with funding from NIH-NCI CCSG: P30 014195, NINDS Neuroscience Core Grant: NS072031 and the Waitt Foundation.

## Author contributions
H.N. and N.K.D. conceived the study and designed experiments. H.N. and C.C. performed experiments. H.N. analyzed the data. H.N. and N.K.D wrote the manuscript. All authors read and accepted the manuscript.

## Additional information

**Competing interests:** The authors declare no competing interests.

