## [Peer Review File · Nature Communications]

Reviewers' Comments:

Reviewer #1:

Remarks to the Author:

Currently, the construction of artificial cell-like structures gains a lot of attention. One of the logical next steps in this context is the establishment of communication and simple multicellular behavior. Previously, several research groups achieved communication between artificial cellular entities via small molecule signaling or via membrane channels. In the present paper, the authors take a different approach by creating artificial cell mimics that consist of a porous polymer membrane containing a "nucleus" comprised of DNA bound to a clay crystal. Notably, the polymer membrane is permeable to large molecules such as proteins, RNA (or the contents of a cell-free TXTL system) and thus communication via larger molecules can be established.

In the paper the authors first characterize their cellular structures and then show that proteins can be expressed and localized (TetR via a tetO array) in the nucleus. Furthermore, they demonstrate a simple form of communication via proteins and also quorum sensing. To this end, a T3 RNA polymerase is expressed and used as a diffusible activator for gene expression.

This is an interesting and important paper for researchers working on cell-free synthetic biology and artificial cellular structures and establishes a novel platform for communication between cell mimics via diffusing proteins. The paper is nicely written and this referee only has minor comments. In a revised version of the manuscript the following points should be considered:

- what is the size/size distribution of the clay crystals – from the images it appears you have exactly one spot in each cell mimic, why is that?
- can you elaborate a little more on the structure of the clay gel? is it a coacervate? why and how is the DNA accessible to TX machinery in the gel?
- other researchers have made hollow spheres using a different technique, e.g., "gel-shell beads" by Hollfelder et al. Nat. Chem. 2014, which may be interesting in this context.
- a similar paper as the Lentini et al., ACS Centr. Sci. 2017, on communication between droplets and bacteria is: Schwarz-Schilling, Integrative Biol. 2016.
- are the cell mimics with the polymer shells mechanically stable?
- it is interesting that the quorum sensing response appears to be sharp, even though there is no positive feedback – why is that? Can this be explained quantitatively (using a simple model)? Also the paper only states in the conclusion that there is no feedback, this should be pointed out and discussed earlier in the text.

Reviewer #2:

Remarks to the Author:

The manuscript by Niederholtmeyer, Chagga, and Devaraj describes the fabrication of a cellular mimic that is capable of synthesising proteins in a structure with sufficient porosity to allow diffusion of the protein from the site of synthesis. These mimics are shown to be capable of cell-cell communication because molecules can freely diffuse out of them and into neighbouring mimics where they can activate gene expression. Overall, the approach is interesting and the results presented are well supported by the data, though I am wondering what kinds of biological questions can really be tackled by such a system. In essence, the question boils down to how well the 'mimics' really do mimic the behaviour of a cell and how the material choices impact the conclusions drawn from the system.

Specific comments:

1. Not being an expert in microfluidics, it was not clear to me what this statement by the authors meant, "...studies on systems that structurally resemble natural cells have focused on single, isolated cell-mimics because of limitations in assembly methods and communication channels." (line 35). Do the authors mean a single type of mimic can only be used because of the way these are fabricated or rather that the physical set up limits the ability to spatially control the communication?
2. The cellular mimics the authors construct are made of non-natural polymers. Does this influence the behaviour of the biological components, e.g. protein or mRNA structures or their interactions? Is the gene expression seen in the Tet or T3 RNAP driven systems in the mimics similar to what would be expected in vivo given the concentrations of species?
3. How does the polymer affect the diffusion rate of the proteins produced? My understanding from the results is that the macromolecules diffuse very freely, which is very different from diffusion in natural biological system. Therefore, is there a mapping that can be used to predict how these changes affect the dynamics of the system so that the behaviour of circuits in the cellular mimics has relevance to a real biological system? Otherwise, the use for prototyping as discussed on Line 265 would be very limited. This would also inhibit the use of such a system for studying developmental processes (Line 270).

Responses to reviewers' comments

We would like to thank the reviewers for their constructive comments and helpful suggestions. In our revised manuscript we have addressed all points raised by the reviewers as detailed below in our direct responses to the individual comments.

Reviewer #1

- what is the size/size distribution of the clay crystals – from the images it appears you have exactly one spot in each cell mimic, why is that?

We thank reviewer #1 for the comments and the interest in the clay-DNA hydrogel nuclei. We are happy to elaborate further on formation and structure of the clay-hydrogel nuclei. We believe that it is indeed an interesting material for cell-free synthetic biology and the construction of biomimetic systems.

To answer the question about the size distribution, we have analyzed sizes of the clay-DNA nuclei in the batch of cell-mimics shown in Figure 1 and Supplementary Fig. 5. We have included these new results in Supplementary Fig. 5. We found that the size distribution of nuclei was slightly broader than the size distribution of the microcapsules surrounding the nuclei. On average, nuclei measured half the size of microcapsules in diameter. We have also added a description of these findings in the main text of the manuscript.

Fig. S5 c) Size distribution of hydrogel nuclei and microcapsules. Areas were measured from GelRed fluorescence and brightfield images for segmented nuclei and microcapsules respectively using Fiji/ImageJ (n = 41). Diameters were calculated from area assuming a circular shape, average and standard deviation are shown. A low percentage (approximately 5 %) of cell-mimics per batch did not contain one defined hydrogel nucleus but fragmented particles.

It is correct that the majority of cell-mimics contained exactly one distinct “nucleus”. Only approximately 5 % of cell-mimics in a given batch did not contain one defined hydrogel nucleus but fragmented particles. Interestingly, the clay-DNA hydrogel formed irregularly shaped and sized particles when hydrogel formation was induced in bulk solution. We explain the regular shape and size of the clay-DNA hydrogel nuclei in cell-mimics with a templating effect of the surrounding microcapsules. Hydrogel formation inside microcapsules is more controlled because polymer membrane shells contain identical amounts of clay minerals and DNA in a confined volume. When

microcapsules are permeabilized, there is probably only limited diffusion of DNA and clay particles out of the microcapsule before the nucleus forms. We have expanded the description of hydrogel nuclei formation and morphologies. Also, we have added a microscopic image of irregularly shaped clay-DNA hydrogel particles formed in a bulk solution for comparison with the regularly shaped nuclei in Supplementary Figure 5.

Fig. S5 a) Morphology of clay-DNA hydrogel particles formed in bulk solution. Microscopy images show a brightfield image (left), fluorescence of the clay-DNA hydrogel stained with GelRed (center) and a merged image (left).

- can you elaborate a little more on the structure of the clay gel? is it a coacervate? why and how is the DNA accessible to TX machinery in the gel?

To answer the question about the structure of the clay hydrogel we have added description and citations about the mechanism of adsorption of nucleic acids to clay minerals, which occurs through electrostatic interactions, cation bridges, hydrogen bonding and ligand exchange (Yang *et al. Sci Rep* **3**, 3165 (2013); Yu *et al. Appl. Clay Sci.* **80-81**, 443–452 (2013), Franchi *et al. Orig. Life Evol. Biosph.* **29**, 297–315 (1999)). DNA in the clay hydrogel is accessible to the transcription machinery because the clay hydrogel is porous and permeable to large macromolecules up to 500 kDa in size. We have expanded the description of these results presented in Supplementary Fig. 3 in the main text of the manuscript. T7, T3 polymerases and the *E. coli* RNA polymerase holoenzyme are smaller than 500 kDa. A previous study cited in the manuscript has found that DNA bound in Laponite XLG clay hydrogel was active in cell-free transcription and translation reactions (Yang *et al. Sci Rep* **3**, 3165 (2013)). Another previous study of clay-DNA complexes contains transmission and scanning electron microscopy images of DNA molecules adsorbed to clay minerals showing that DNA strands are not bound to clay surfaces by their complete lengths and therefore at least partly accessible (Franchi *et al. Orig. Life Evol. Biosph.* **29**, 297–315 (1999)).

- other researchers have made hollow spheres using a different technique, e.g., “gel-shell beads” by Hollfelder *et al. Nat. Chem.* 2014, which may be interesting in this context.

We thank the reviewer for this suggestion. We have added the suggested reference and expanded the section to other previous works on hollow, porous spheres (like gel-shell beads, polymersomes, proteinosomes and colloidosomes) in the introduction.

- a similar paper as the Lentini et al., ACS Centr. Sci. 2017, on communication between droplets and bacteria is: Schwarz-Schilling, Integrative Biol. 2016.

The citation was added at relevant positions in the introduction.

- are the cell mimics with the polymer shells mechanically stable?

While we have not quantitatively measured the mechanical properties of the polymer shells, we found that the polymer membranes of cell-mimics were highly stable to standard handling in the lab (e.g. centrifugation at high speeds, pipetting, chemicals and solvents). Polymer membranes of cell-mimics did not deform when flowed in microfluidic channels or during other microscopic analyses in liquids. The rigidity of the membranes can be judged from the SEM image of a dried microcapsule (Supplementary Fig. 2a). Upon drying, dents formed in polymer membranes but they did not completely collapse. Polymer shells could be broken using a razor blade (Supplementary Fig. 2c). We have expanded the description of the observed mechanical properties and added a reference to a study that performed quantitative mechanical measurements on porous microcapsules formed in a comparable process and with a similar polymer (Loiseau *et al. Langmuir* **33**, 2402–2410 (2017)).

- it is interesting that the quorum sensing response appears to be sharp, even though there is no positive feedback – why is that? Can this be explained quantitatively (using a simple model)? Also the paper only states in the conclusion that there is no feedback, this should be pointed out and discussed earlier in the text.

We agree that the sharpness of the artificial quorum sensing response is interesting. Partly it can be explained by the response curve of the T3 promoter as a function of T3 RNAP template concentration in bulk solution (Supplementary Fig. 14), which had a sigmoidal shape with a half maximal induction at a concentration that corresponds to the calculated T3 RNAP template concentration with cell-mimics at the switching density. We have added indicators to corresponding cell-mimic densities to Supplementary Fig. 14. As can be seen from Supplementary Fig. 14, the sigmoidal shape of the pT3 response curve is not sufficient to explain entirely how sharp the response to density was in cell-mimics. Ultrasensitive, switch-like responses can arise from other molecular mechanisms than positive feedback loops that cause natural quorum sensing responses. We hypothesize that the sharp switch in gene expression in response to cell-mimic density may be due to the high valency of hydrogel nuclei containing local T3 promoter concentrations of 200 nM, combined with large distances between cell-mimics at lower densities. Mathematical modeling has suggested that cooperative binding can occur between polyvalent ligands and single receptor sites (Klein, Pawson, & Tyers, *Curr. Biol.* **13**, 1669–1678 (2003)), which may be translatable to hydrogel nuclei and T3 RNAP in our case. We estimate that on average a hydrogel nucleus contains approximately 200,000 T3 RNAP promoters, making it extremely polyvalent. This suggests that once a T3 RNAP encounters a nucleus, it will likely not diffuse away after transcribing a reporter

gene but bind to another T3 promoter in its proximity. However, when cell-mimics are at low density, they only produce a low concentration of T3 RNAP and between cell-mimics there are large distances that a free T3 RNAP must explore by diffusion before encountering a hydrogel nucleus. At high densities of cell-mimics T3 RNAP concentrations are higher and distances are shorter, making encounters much more likely. To illustrate this proposed mechanism of the sharp quorum sensing response we have added Supplementary Fig. 16 and expanded the discussion to include our qualitative model of the proposed mechanism, as well as references supporting our hypothesis. A quantitative model should include affinities, synthesis and diffusion rates and is beyond the scope of this study.

Fig. S16. Proposed mechanism for switch-like response of gene expression to cell-mimic density. a) Binding of T3 RNAP to polyvalent hydrogel nuclei of cell-mimics. Schematic adapted from Klein, Pawson, & Tyers, *Curr. Biol.* **13**, 1669–1678 (2003). b) Schematic of cell-mimic and T3 RNAP distributions at low and high cell-mimic densities.

As suggested, we have added a comment about the absence of feedback loops in our quorum sensing network in the results section as we agree that this is a noteworthy difference to natural quorum sensing mechanisms.

Reviewer #2

1. Not being an expert in microfluidics, it was not clear to me what this statement by the authors meant, "...studies on systems that structurally resemble natural cells have focused on single, isolated cell-mimics because of limitations in assembly methods and communication channels." (line 35). Do the authors mean a single type of mimic can only be used because of the way these are fabricated or rather that the physical set up limits the ability to spatially control the communication?

We thank reviewer #2 for the comments. What we wanted to point out was the fact that previous man-made systems, which resembled natural cells in that they encapsulated biomolecules in semi-permeable membranes, could often not be produced reliably and in large quantities (recent microfluidic methods have begun to solve this problem and are cited), and that communication was limited to few and only small signaling molecules. We agree that the wording was unclear and have clarified the section.

2. The cellular mimics the authors construct are made of non-natural polymers. Does this influence the behaviour of the biological components, e.g. protein or mRNA structures or their interactions? Is the gene expression seen in the Tet or T3 RNAP driven systems in the mimics similar to what would be expected in vivo given the concentrations of species?

We agree with reviewer #2 that our cell-mimics, with their polymeric membranes, are a highly artificial system. To give an example for the question about the influence on biological components, we point out in the manuscript that we found it necessary to PEGylate the highly, porous polymer membranes to prevent non-specific protein binding (see also Supplementary Fig. 6). This example illustrates that indeed the use of non-natural materials can influence the behavior of biological components but that engineering solutions may be found. In general, we do not expect protein and mRNA structures and interactions to be greatly altered in our system. After all, the *E. coli* transcription and translation machinery was functional and produced functional proteins. Cell-free TX-TL systems have been shown to work not only in test tubes but also after freeze-drying, on paper, in emulsions, vesicles and microfluidic devices. However, as pointed out, large surface areas or specific material properties can influence expression yields.

DNA template concentrations inside cell-mimics were comparable to concentrations of low to medium copy number plasmids in *E. coli* (see Supplementary Table 2). Highly expressed proteins reach low μM concentrations inside cells, which is comparable to the concentrations TX-TL systems produce (see for example Supplementary Fig. 14 for data from this study and Moran, Phillips, & Milo, *Cell* **141**, 1262–1262.e1 (2010)). With respect to comparisons to cells, we would like to point out that we were not aiming to create a system that works exactly like a cell. Our goal was a synthetic, hybrid system assembled from both biological and non-natural building blocks that emulates some features of natural cells. Because of this we do not directly compare gene expression and concentrations between cell-mimics and natural cells in our manuscript. Addressing also point number 3 of the reviewer, we have changed the wording from “model” to “emulate” throughout the manuscript, and clarified in the discussion that we were not aiming to directly model processes in natural cells (see below).

3. How does the polymer affect the diffusion rate of the proteins produced? My understanding from the results is that the macromolecules diffuse very freely, which is very different from diffusion in natural biological system. Therefore, is there a mapping that can be used to predict how these changes affect the dynamics of the system so that the behaviour of circuits in the cellular mimics has relevance to a real biological system? Otherwise, the use for prototyping as discussed on Line 265 would be very limited. This would also inhibit the use of such a system for studying developmental processes (Line 270).

The reviewer raises an important point about the diffusion rates of proteins across polymer membranes and between cell-mimics. It is correct that

proteins diffuse freely between cell-mimics unless they are targeted to hydrogel nuclei by binding sites on the DNA (see Supplementary Fig. 10). In future work, diffusion rates of proteins may be tuned by modifying membrane permeability or using nuclear binding sites that restrict diffusion of DNA binding proteins. We have added this point to the discussion. Additionally, diffusion rates might not necessarily need to be the same as in natural biological systems to test reaction-diffusion models and emulate pattern formation in development, instead they could be tested on different scales and using different molecular mechanisms. To clarify this, we rewrote the sentence in line 270 to read “We believe that reaction-diffusion models explaining developmental processes in multicellular organisms may be tested and emulated in artificial, tissue-like arrangements of cell-mimics even when they are composed of unnatural materials, operate on different scales and with different molecular mechanisms”. Furthermore, as mentioned above, we replaced the word “model” with “emulate” throughout the manuscript to clarify that we do not propose “direct modeling” of natural systems.

We agree that “prototyping” in line 265 was misleading. We did not mean prototyping for the implementation in natural systems. We changed the sentence to “Such modularity might facilitate the assembly of functional, distributed gene circuits by titrating cell-mimics containing different parts of a network”.

Reviewers' Comments:

Reviewer #1:

Remarks to the Author:

The authors have responded well to all the reviewer comments and improved their manuscript further. This reviewer has no further comments and the manuscript may be published as is.

Reviewer #2:

Remarks to the Author:

Overall, the reviewers have addressed all comments by myself and the other reviewer to my satisfaction. I have no additional queries.